# Multimodal Imagery in Forensic Incident Scene Documentation

**DOI:** 10.3390/s21041407

**Published:** 2021-02-17

**Authors:** Leszek Luchowski, Dariusz Pojda, Agnieszka Anna Tomaka, Krzysztof Skabek, Przemysław Kowalski

**Affiliations:** 1Institute of Theoretical and Applied Informatics, Polish Academy of Sciences, Bałtycka 5, 44-100 Gliwice, Poland; dpojda@iitis.pl (D.P.); ines@iitis.pl (A.A.T.); 2Institute of Computer Science, Cracow University of Technology, Warszawska 24, 31-155 Kraków, Poland; kskabek@pk.edu.pl; 3KiperTech Consulting, Szwedzka 52, 30-315 Kraków, Poland; prko@kipertech.pl

**Keywords:** crime scene documentation, forensic science, 3D imaging, large scale data handling

## Abstract

Various imaging modalities are evaluated for use in forensic incident (crime or accident) scene documentation. Particular attention is paid to the precision vs. cost tradeoff, accomplished by judiciously combining various 3D scans and photogrammetric reconstructions from 2D photographs. Assumptions are proposed for two complementary software systems: an event scene pilot assisting the on-site staff in their work securing evidence and facilitating their communication with stationary support staff, and an evidence keeper, managing the voluminous and varied database of accumulated imagery, textual notes and physical evidence inventory.

## 1. Introduction

Investigators dispatched to the scene of a crime or an accident will inspect, and possibly collect, any evidence that can help identify the causes, reconstruct the course of events, and prosecute any perpetrators. They need to collect as much evidence as possible, often in a limited time, under significant stress, and in harsh conditions.

Modern digital technology offers many advanced means of acquiring, storing, transmitting, sharing, and processing information, often in a multimedial format. These new digitization techniques can replace the traditional means of collecting evidence, or supplement them by acquiring information which has so far been impossible to obtain.

It is also important to distinguish the measurements which can be carried out at the scene from those which need to be referred to the forensic lab or other controlled venue.

The total information collected at the scene, from fingerprints to 3D scans of artifacts, is highly varied in kind and bulky in size (up to the order of terabytes). Collecting it requires judicious choice of objects of interest, imaging devices, their positions, and parameters. Field technicians must react flexibly to the situation on site, but they also need support from an operations center where support staff working in more comfortable conditions and with better computer infrastructure can process the data, add geographic coordinates, relate the various imaging modalities to each other, and make decisions on what additional images need to be taken.

### 1.1. Purpose of the Present Work

The purpose of this paper was to review imaging and other data collection techniques applicable at the incident scene, and to propose an approach to integrating the collected multimodal data in a manageable format convenient for the forensic expert to use, both at the field while collecting the evidence and at the lab while supporting the field staff and post-processing the evidence. Specific properties of each imaging modality and the different data formats must be taken into consideration so that all the imagery can be integrated, ordered, and arranged into a hierarchy according to level of detail and to relevance. In particular, high-resolution local scans of objects of interest need to be incorporated into overview scans of the entire scene. Regardless of the progress of scanning technology, overview scans are likely to remain at a relatively low resolution, in order to keep the data volume workable. At the same time, the way objects of interest are represented in them can only be used as an indication of their position, and is highly inadequate for any examination of their appearance or state. While detailed scans of objects are valuable in themselves, embedding them in the overview scan can provide additional insight into the course of events which resulted in both the observed location and condition of objects. Photogrammetric reconstruction is much cheaper and more convenient than laser scanning; one of the purposes of the present work was to verify if its precision is sufficient to warrant using it as an alternative.

Geometric integration also allows multiple imaging modalities (various types of scanners, photogrammetric reconstructions) to be presented in one scene model.

The choice of level of detail is not binary. Most scanners are either built for long-range or close-range scanning, but photographic images can be taken at any distance and zoom setting, allowing the range and resolution of photogrammetric reconstructions to be tuned to the investigators’ needs. Currently existing photogrammetry software only accepts, for a single reconstruction, photographs taken at a similar scale. If photographs of radically different scale are to be used jointly, without intermediate views, the close-up reconstruction has to be manually edited into a wider one (Figure 1).

It is hoped that the hierarchic approach to scene modeling and the idea of two interconnected application programs managing on-scene work and evidence storage, presented in this paper, will be helpful to the development of forensic incident-scene documentation systems.

### 1.2. State-of-the-Art

Nearly all 3D scanner manufacturers and dedicated software developers declare that their products can be used in forensics. Their use for such purposes has also been mentioned in literature [1,2]. Most software supplied with scanners uses proprietary data formats, incompatible with each other. Authors mention single devices and measurement techniques [3,4] as well as, increasingly often, review and compare various devices in terms of their capabilities, their compliance with the requirements of field work, their accuracy, and cost to value ratio [5,6]. Capabilities being offered include documenting the scene and creating visualizations of measurements to be presented in court, as well as simplified measurement-based modeling to perform various kinds of simulations. These simulations, also called forensic reconstruction, which were unavailable using traditional documentation methods, are possible in a 3D virtual reality [1]. Forensic reconstruction in this sense is not the subject of the present research endeavor. It is a very delicate process, as signaled in [7], because not all circumstances can be inferred from the evidence, and a visualization will inevitably only show one of the many possible appearances, locations, and event chronologies.

### 1.3. The Structure of the Paper

The paper presents a brief description of multimedial techniques of digital documentation of the forensics scene, regarding their accuracy and specific factors influencing the integration techniques. The perspectives of using photogrammetry as an economic substitute for 3D scanning are discussed. Assumptions are proposed for two application programs: one facilitating the coordination of on-site evidence collection, which will be referred to as the incident scene pilot, and another—the evidence keeper—managing the collected data in digital storage, also assisting in its integration.

## 2. Imaging Modalities and Data Formats

While only expert opinions are admissible in court, they can be based on, and supported with, the digitized information.

To evaluate this information, the expert needs to know its accuracy and credibility, i.e., the way it was created and the full information about the chain of processing from the scanner or camera to the numeric data, images, and visualizations delivered to his screen or desk. Like the chain of custody associated with physical evidence, this chain of processing allows the expert to make informed judgments about the origin, format, relevance, integrity, and accuracy of the information.

### 2.1. Manual Sketch and Description

To evaluate the new 3D optical measurement techniques, one should relate them to the traditional techniques of crime scene documentation, including taking sketches by hand on millimeter paper, using a tape measure for distances and sizes, and taking 2D photos of the general overview of the scene and of selected details. As most of those methods can be described as manual, the human factor plays an important role in deciding what should be measured, affecting measurement precision, introducing subjective error, and being limited in the access to various points of the scene. Moreover, measurements can only be made during on-scene work, and cannot be repeated at a later date.

Apart from being moved to a digital medium, these modalities can remain essentially unchanged from the traditional paper-and-pencil work of a crime scene technician. The resulting files weigh tens of kilobytes at most and do not pose a problem in storage and transmission.

One digital development which can improve both the speed and the accuracy of sketch taking is the automatic generation of preliminary sketches from 3D scans. It will be discussed below in Section 2.

### 2.2. Photographs—2D Stills and Videos

Photography has long been used to document crime scenes. Due to the relatively low cost, its use is very popular, large areas of the scene are covered, and the choice of photos to be used can done in the lab.

Some jurisdictions may require forensic digital photographs to be stored in a lossless format such as TIFF. With video footage, storage and transmission may become an issue, depending on the duration and the settings of the camera, and especially when many cameras are used.

A special kind of 2D picture is the spherical panoramas created by terrestrial laser scanners. Collected automatically during scanning, centered at strategic locations, they cover nearly the whole view sphere at a high resolution. However, they can be difficult to correctly display in court without specialized immersive viewing software (or, ideally, hardware).

### 2.3. Photogrammetry

Photogrammetry [8,9,10,11] is a technique of reconstructing a 3D scene using its photographs taken from multiple viewpoints. The photogrammetric approach to forensic documentation was tested in several papers [12] and proved its great potential, even if the scenes were contaminated [13].

The algorithms use the principles of stereovision [14], determining the distance to a point from two or more photographs. Unlike classical stereovision, the projection matrices of cameras are not known in advance, but are determined during the reconstruction. Given more than two views of a static scene, the redundancy of geometric information allows the 3D coordinates and the projection matrices to be reconstructed in the same process.

In practice, sequences of up to hundreds of photographs from multiple viewpoints are used. For correct reconstruction, photographs from neighboring viewpoints should overlap. As photogrammetry alone only recovers shapes up to a scale factor, a calibration object of known dimensions must be present in the scene if quantitative measurements are required. Of the various available implementations, the Agisoft Metashape software [15] was used in our tests.

The photogrammetric software tools currently available follow a number of processing stages: the reconstruction of easily identifiable feature points, generating depth maps for the individual cameras (camera positions), increasing the density of reconstructed 3D points, and reconstructing a triangular mesh interpolating the object surface. The precision of photogrammetric reconstruction strongly depends on the quality of photographs, i.e., the lighting conditions, image focus, and resolution, as well as on the distance of the camera(s) from the scene. At the same camera resolution, a different quality of reconstruction will be obtained from general views than from detail close-ups. Figure 2 and successive ones show a series of photographs used for photogrammetric reconstruction. The photographs are virtually embedded in the 3D space of the scene, placed at locations they were taken from.

Photogrammetric reconstruction can use purpose-made photographs as well as pictures of the scene taken for general documentation purposes.

The images need to be processed by complex algorithms, which may require operations-center level hardware. The raw data may total hundreds of megabytes; the processing requires tens of gigabytes of intermediate storage; and the resulting model is in the order of up to single gigabytes. In most situations, this means the images need to be transferred to the lab for processing, and then either the mesh or, more probably, its projected 2D views can be transferred back as needed For very large volumes of data, the transfer time (over the cellular digital network) may take considerable time, especially in remote areas where coverage is limited.

### 2.4. 3D Scanning

3D scanners are devices measuring the distance to points in a given scene. According to measurement method, they can be divided into [16]:scanners using the principle of triangulation, which is the basis of stereoscopy and structured-light scanning. All triangulating 3D scanners mentioned in this work are optical, i.e., using visible light, or in some cases infrared, to acquire a 3D model of surfaces of material objects. Typically, the data from a triangulating scanner is represented by a triangular mesh.Scanners using a point laser rangefinder, based either on time of flight or phase shift, and a deflection unit sweeping the beam across space. This class of scanners is known as Terrestrial Laser Scanners (TLS). By rotating the beam in two angular dimensions (typically azimuth and elevation), the space around the scanner is sampled, and at each sample point the distance is measured to the nearest surface encountered by the beam. The result is a point cloud in a spherical coordinate system. Most scanners also collect regular RGB data known as texture, like ordinary photographic cameras, as well as reflective indices. As TLS devices cover nearly the full 4π steradians solid angle, the resulting photograph is a wide angle panorama in spherical coordinates. Certain devices may acquire other types of point information, such as temperature.

From the point of view of the intended class of applications—and the required accuracy and range—scanners can be divided into long-distance scanners (mostly TLS) creating an overview of the layout of the scene, and precision scanners intended for smaller objects (often, though not always, triangulating).

Like in photogrammetry, building a complete model of the scene requires image data from multiple viewpoints to be integrated. Some triangulating scanners are handheld, meant to be moved over the object to collect multiple scans.

Scanners are legally allowed for forensic applications [17], and such usage has been described in [1,2,18,19].

3D scanning is by far the most data-intensive imaging modality. Volumes of data up to the order of terabytes are often impossible to transmit from the scene to the support center, so they have to be stored on disks in the technician’s equipment. Reduced sets of data, such as panoramic photos and low-resolution depth maps, can be transmitted to keep the support center informed about the progress of scanning.

### 2.5. Collecting Physical Samples and Artifacts

Collecting material for analysis in the lab should, if possible, be done after the digitization of the scene, and after marking the location of the material in sketches and taking in situ photographs of them.

### 2.6. GNSS Positioning Systems

In outdoor incident scenes, especially where there are few buildings or other permanent reference points, the geographic location of pieces of evidence and of the entire scene can be documented using satellite navigation systems.

The accuracy of the civilian GPS system without support is a few meters, inferior by a factor of 10 to the military version. Greater accuracy can be achieved with augmentation systems. Using Wide Area Augmentation System using satellites (WAAS) and Local Area Augmentation System (LAAS, with terrestrial stations) an accuracy of approx. 1 m can be obtained. The required accuracy is obtained by professional GNSS systems used by surveyors thanks to the use of differential measurement technology, such as static Differential Global Positioning System (DGPS) [20]. Techniques similar to DGPS can also be used for other GNSS systems, like GLONASS. DGPS uses difference between measured satellite signals, and signals for a known, fixed position. It should be noted that professional GNSS positioning devices can use data from different GNSS systems, e.g., GPS, GLONASS, or Galileo, in a way that is transparent to the user.

## 3. Accuracy

Various constructs, under various names—precision, accuracy, trueness—are used to gauge the quality of a measurement. The present work discusses precision, i.e., the repeatability of measurements, and accuracy, i.e., closeness to the actual physical quantity or a reference measurement. The international standard [21] lays out a complex and exhaustive procedure for the assessment of measuring device precision and accuracy. The present work does not aim to undertake such comprehensive experiments, but to give a simple working estimate of what can be expected from the various 3D imaging techniques.

### 3.1. Methods of Determining Accuracy

Choosing the method of comparing the accuracy of measurement devices is no trivial task, as the devices differ in measurement technique and in the way they represent data. The most intuitive method of weighing the merits of various 3D imaging devices and techniques is to compare the linear distances between points in each image. This verification technique can be used in a medical imaging context [22], and for forensic purposes [23]. This approach assumes that there is a known correspondence between features in the two datasets, but their alignment is not obligatory. The idea has been extended to analyzing the matrix of all possible distances between the localized features; this approach is known as Euclidean Distance Matrix Analysis (EDMA) [24]. Most often, the homology information is provided by a human expert via a man–machine interface which allows him to pinpoint features and label them. The precision obtained in this way is a combination of human skill and device quality.

However, the correspondence of features in images cannot always be provided by a human operator. When such input is not available, the idea of temporary correspondence used in the Iterative Closest Point (ICP) algorithm can be applied [25]. It is a registration process in which, at any given stage, points in one image are matched to their closest neighbors from the other. As the matching progresses, the two images are moved against each other and point correspondences change. The registration is prerequisite for the determination of the correspondence, which allows the EDMA to compare analogous point-to-point distances in either image. The method was originally developed for small matrices, and its application is limited by the volume of the distance matrix. Table 1 shows the results of applying this technique to a small fragment (Figure 3a) of the reference object.

The ICP algorithm aligns sets of points—called clouds of points in 3D imaging terminology—to minimize the distance between those sets. The distance between clouds is calculated as the average of distances between points from one set and their counterparts (the closest ones) from the other. Treating one of the cloud as the reference, the average distance between corresponding points can be regarded as the accuracy, and the standard deviation as the precision of measurement. Exemplary results of Cloud to Cloud (C2C) comparison are shown in Table 1.

In some cases the cloud points are the only information available for comparison. The C2C method suffers from serious limitations due to the discrete scanning techniques of 3D scanners. The sampling resulting from the rotation of the measuring unit or from finite camera resolution makes the resolution of the cloud highly dependent on the type of device used, its settings (angular step, zoom, etc.), and the distance to the target. As shown in Figure 3b, the vertices of red and blue meshes on the surface scanned from different scanner positions do not correspond to each other as it is assumed in the C2C comparison method.

Some 3D scanners yield a polygonal mesh which interpolates the scanned surface. When at least one of the structures being compared has a known surface, correspondences can be found by seeking, for each point of the opposite cloud, the closest point on this surface (Cloud to Mesh, C2M). The same procedure can be applied for Mesh-to-Mesh comparison, seeking the correspondences for each vertex of one mesh on the surface of the other.

In order to find the average distance between surfaces *S* and $S^{\prime }$ the formula given by [26] can be used:(1)$$E_{m}\left(S,S^{\prime }\right)=\frac{1}{\left|S\right|}\int_{S}\min_{p^{\prime }\in S^{\prime }}\left(d(p,p^{\prime })\right)ds$$
where for each vertex *p* of *S* the closest point $p^{\prime }$ on $S^{\prime }$ is estimated, *d* is the distance measured between *p* and $p^{\prime }$, and $E_{m}$ is the distance error. For the mesh representation the distance error can be calculated as the average of the distances measured for all vertices of the mesh *S*:(2)$$E_{m}\left(S,S^{\prime }\right)=\frac{1}{N}\sum_{i=1}^{N}d(p_{i},p_{i}^{\prime })$$The results of exemplary usage of the method are shown in the Table 1.

The euclidean distance matrix analysis approach has significant potential for assessing the differences between shapes as seen by different scanners. However, it also has limitations because it uses an N∗N matrix, with *N* being the number of points; this means quadratic computational complexity over datasets that are usually very voluminous.

The methods which directly compare distances between points of different scans are biased by the error of point matching, as points in different scans do not coincide even if they represent the same surface.

Table 1 shows that a smaller average error is obtained with point-to-surface distances, even though an interpolation step is involved. For this reason, the C2M method was adopted for further analysis.

### 3.2. The Accuracy and Precision of Measuring Devices

To integrate the data, we should estimate the accuracy with which the data from various devices can be brought into a single coordinate system. While the devices have nominal accuracies declared by their manufacturers, the actual accuracy was estimated experimentally and is presented below.

The quality of a measuring device—its accuracy and precision—can be assessed by taking a sequence of measurements of a reference object of known dimensions [27,28] under the same set of conditions. By changing those conditions, the sensitivity of measurements to them can be estimated.

The present work is based on reusing data from previous research, without a possibility to engage in new experiments. However, in many of the available datasets, a dummy head (always the same one) is present in the scene (Figure 2 and Figure 4). Unfortunately, its position relative to the imaging device is different every time; other imaging conditions were also not repetitive or fully controlled. The results presented here will thus be an overview of the accuracy and precision of devices rather than a full and regular analysis. As absolute information about the dummy head is not available, high-quality scans taken with a Konica Minolta VI-9i equipped with tele lens were used as reference. The scanner has a nominal accuracy of 0.05 mm and a resolution, at the distance used, of approximately 1200 points per cm^2^.

Some of the test scans had been obtained using the same Minolta scanner with the middle lens. The Faro X 130 and Z+F 5010C scanners had been used in their two available modes: overall scan and detail scan.

From each 3D scan or reconstruction in which the dummy head was present, the fragment representing this head was manually selected, discarding the rest of the data (Figure 5). The measurements were taken in the following way: the selected parts of 3D scans or photogrammetric reconstructions were registered to the reference model. Then the distance was measured from each vertex of the scan or reconstruction to the surface of the reference model. The mean distance and standard deviation were determined. The resolution of the scans was estimated by dividing the number of points in the scan by the surface area of the mesh built on those points.

Table 2 shows the estimated accuracy, precision, and resolution of the photogrammetric 3D reconstructions for the case where the camera was placed close to the dummy head, or at a distance where it could also see neighboring objects, and for the case of overview photographs of the entire outdoor scene.

Table 3 presents the accuracy estimates for different close-range 3D scanners and TLS scanners, based only on the limited data used in this research (Figure 5).

To summarize the results presented in the tables, it can be said that the accuracy and precision of scanning/reconstruction depends on the type of device used, its settings (angular step, zoom, etc.), and the distance to the target. The resulting resolution depends not only on the nominal angular resolution of the scanner but also on other parameters, target distance, and the scanning techniques (incremental scanning with the Faro Arm).

For close-range scanners, the estimated accuracy is about 0.05 to 0.5 mm, and the resolution can range from tens to a thousand points per cm^2^. For long-range scanners, the estimated accuracy is approximate 0.4 to 1.5 mm, and their surface resolution is 3–150 points per cm^2^. At less than ten points per cm^2^ the surface is poorly rendered in spite of high measurement precision. However, increased resolution does not always ensure an improved quality of the scan of the dummy face. For dense point clouds scanned with relatively poor precision, the resulting surface is very noisy.

Thus, long-range scanners are suitable for overview scans of the entire scene, but close-up scans of objects should be done with close-range scanners delivering a better resolution and a better rendering of surfaces. Long-range 3D imagery, with a fine close-up 3D model embedded in them can be an example of cross-scale integration of 3D representations. Such a combined multiscale model may be delivered to the investigator as a virtual environment to be browsed and progressively zoomed into as the need arises.

The estimated accuracy of 3D photogrammetric reconstructions is often inferior to that of 3D scanners, and error can reach 9 mm when using photographs taken at a large distance. However, using multiple close-up photos in high resolution, an accuracy of fractions of a millimeter—about 0.2 mm can be achieved—which is comparable to close-range scanners. Considering the lower cost of photographic cameras, photogrammetry may in the future become an alternative to scanning. From a forensic point of view, photogrammetry has the advantage of reconstructing the positions of the camera, which can be used to better illustrate the documentation in court.

### 3.3. What Influences the Precision of Photogrammetry?

The quality of the photogrammetric reconstruction depends on many factors that can be classified into several groups [9]: the quality of the camera and the photos taken, properties of the object, the selection of reconstruction parameters.

Camera quality parameters include image resolution and the quality and repeatability of the internal parameters of the lens. The method of taking photographs involves fixed values of focal length and brightness, the method of picture cropping, distance from the scene, size of the overlap between adjacent pictures, the presence of markers enabling the scale to be determined, the diversity of structure and color of the object, and dispersed lighting. Reconstruction parameters include: the number of key points in the images, elimination of less significant points, the chosen method of dense map creation, and the strength of point decimation for meshing.

When the photogrammetric reconstruction is compared to the laser scan, the distance to the target should be taken into account, because it has a significant impact on the accuracy of either technique, and also, in the case of laser scanning, on the sampling density. Single laser scans usually have an even spherical distribution of sampling directions. In the case of photogrammetric reconstruction, the density depends on the successfully matched feature points in the images. The greater the number of reconstructed points, the more varied in structure and color the reconstruction will be.

### 3.4. Practical Tests of Photogrammetric Methods

Figure 6 shows the superposition of the laser scans and the relevant photogrammetric reconstructions for selected fragments of the real scene. Both in the case of the laser scans and the photogrammetric reconstructions, there is a problem of fitting the surface components together. Laser scans are obtained from different observation positions. An aligning procedure yields superimposed point clouds, which are then merged into a redundant point structure of increased density. The aligning procedure is decisive for the resulting surface quality. Unfortunately, the nominal measurement accuracy of laser scanners does not take into account such modifications of the measured points.

Photogrammetric reconstruction delivers a complete cloud of reconstructed 3D points. However, to compare it with other surfaces, an aligning procedure is also needed. Figure 7 shows 3D images from different sources aligned and compared. Moreover, the aligning should take into account the change of scale, because, even using size markers, the size of the object may be variable and requires additional correction.

For the example series of image data, the influence of the angular distance between two consecutive images on the number of key points was determined. Calculations were carried out for 6 series of measurements. The photographs were taken using a free-hand camera. The total angular extent of the measurements for all series was determined in a plane parallel to the ground (Figure 8a), and it reached 127°. On this basis the angle of a single measurement step was estimated as the mean value: 4.53°. The results were similar for all series so the average measurement step was assumed as 5°.

Influence of angular distance of observations on the number of reconstructed key-points was evaluated (Figure 8b).

It was clear that if the distance is shorter the number of reconstructed key-points increases. However, this is not always the case, as for particular series the number of reconstructed points for the angular distance $5^{\circ }$ decreased. It is significant that for angular distances greater than $30^{\circ }$, the number of reconstructed points tends to zero.

The influence of the number of input images on the quality of reconstruction was also tested. Quality was understood as both the number of reconstructed key-points (Figure 9a) and the area of the resulting mesh (Figure 9b). Each of these parameters was plotted against the number of input images. This simple analysis showed a linear character of both dependencies. The $R^{2}$ value for linear regression was approximately 0.95, confirming a good linearity of the data.

## 4. Managing and Integrating the Evidence Data

### 4.1. Delivering the Imagery Data to the User

The information contained in a 3D scan can be made accessible to investigators in at least two ways:As a 3D model displayed in a 3D graphic program.As a 2D sketch (usually a ground/floor plan) similar to those taken by hand, but more precise in its dimensions.

We examined the visualization possibilities offered by the software provided by scanner manufacturers, and implemented our own prototype program, IncidentViewer, to experiment with various approaches to processing and presentation.
While testing scanners by Z+F and Faro, we also had the opportunity to use their corresponding software packages: Z+F Laser Control and FARO Scene. Both programs present the data in two forms that are natural to scanners: as a 2D table of distances (depth map) and as a 3D visualization, allowing the user to virtually move around in the scene, and to generate its projections such as floor plans.Figure 10 shows a depth map and a map of reflective indices.Using a 3D scan to automatically generate a sketch: 3D scans can be used to generate a skeleton sketch, which will help the technician who makes a sketch of the crime scene manually. The technician will interpret and annotate the skeleton sketch, attributing lines and spots to physical objects.Another use of automatically generated skeleton sketches is to support investigators operating at the scene. A person supervising these activities from the support center can indicate to technicians the locations that require additional attention.The Z+F software generates sketches by intersecting a 3D scan with several horizontal planes and integrating the cross-sections into one sketch. This approach seems logical, but it has some disadvantages:
it requires a decision about the level(s) of the cross-section(s). The sketches can only be modified or complemented with additional details by adding more cross-sections. Assuming that a sketch is made at the scene (not in the lab), some details may be omitted, and adding them later on may jeopardize its admissibility in court;some cross-sections may need rescaling.

**Figure 10 sensors-21-01407-f010:**
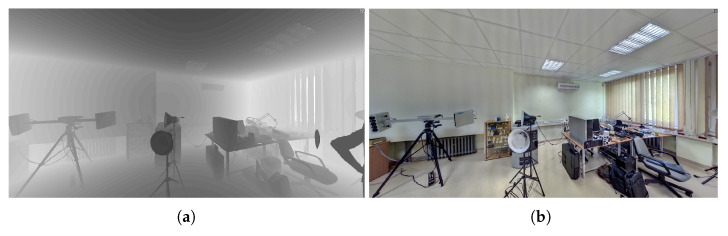
An indoor scene scanned by Z+F 5010C and displayed as depth map (**a**), darker=closer) and reflective index (**b**).

The method proposed by the present work (and, to the knowledge of the authors, previously unpublished) is to detect edges in the panoramic photograph associated with a scan, then project them from the 3D model onto a horizontal plane. The projection has the properties of a sketch (Figure 11) and will only require a technician to identify, highlight, and annotate relevant elements. Moreover, the sketch can be automatically dimensioned by using the 3D data. While algorithms exist to detect edges in a 3D structure, processing the amount of data involved may be unfeasible at the scene. Edge detection in a 2D image is faster by a few orders of magnitude. Moreover, edges detected in the 2D panoramic image can be used to filter away the inessential (non-edge) points from the 3D scan.

### 4.2. Integration of Images from a Single Modality

Almost every imaging modality has its matching application program that controls the device, performs some early processing, and exports the resulting images or other data from a proprietary format to a shared standard.

Integrating data from multiple scans means converting them to a common coordinate system and bringing them into register (aligning). After this process, known as registration, the positions of the imaging devices are described in the common coordinate system, and a joint model of the object or scene is created by merging the representations. Scanner manufacturers provide such functions in their dedicated software.

One example of such software is Z+F Laser Control, an application program dedicated to Z+F scanners, which facilitates the scanning and also allows multiple scans from various viewpoints to be integrated. The viewpoints are identified in the common coordinate system and marked on a map of the scene, and a panoramic view from a selected viewpoint can be displayed. A connection between the panorama and a 3D view allows the user to define a region of interest (Figure 12). A viewing mode integrating all the data helps identify missing fragments of the scene. Measurements are also implemented.

As photogrammetry performs both 3D scene reconstruction and auto-calibration of the cameras, the results can be shown in a map as presented in Figure 13, where a circle of camera positions surrounds the reconstructed scene, with images of the scene as viewed from each viewpoint. A projection matrix and a depth map is determined for each camera.

### 4.3. Integrating the Modalities

3D images taken with a long- or medium-range scanner represent the overall layout of the scene: the positions of buildings, vehicles, furniture, bodies, etc. 3D reconstruction of the whole scene provides information about the position of objects in the scene, but the objects themselves is represented by a small number of mesh points and with poor precision. It is therefore important to make a series of close-up pictures. Details and small objects judged important for the investigation are scanned with short-range (often hand-held) scanners. This leads to a hierarchic approach to the representation and analysis of the scene.

2D photographs, hand drawings of the scene, 3D scans in the form of point clouds, containing 3D coordinates as well as data about reflectance and color; 3D surface reconstructions in the form of meshes, as well as processing results are all types of data which can be acquired from the real-world scene of a crime or accident. Each of these types can be expressed in its own particular device-dependent format and can be displayed in a dedicated application; however, from a functional point of view, all the images involved fall into two categories: 2D and 3D. Handling the entire set of imagery is no small challenge. The sheer volume of the data approaches the limits of standard computers, both in the case of 3D scans and 2D panoramas delivered by scanners (up to hundreds of megapixels).

Quite often, device-dedicated application programs are not able to read image data acquired by devices from other manufacturers. External software such as the evidence keeper has to mediate between the various proprietary formats, reading and processing data from the various 2D and 3D modalities.

Integrating the information is the task of the specialist, and must be facilitated by the evidence keeper software managing this considerable volume of multimodal data. The simplest solution would be to place all the data in a container, only connected by the scene number. It is also possible to relate the images to each other geometrically, forming a virtual model of the scene in a common coordinate system situated in a physical map of the scene location. Other forensic data can be referred to the same map and model.

In general, while a single scan is usually expressed in scanner-centered coordinates, a merged model uses a selected common coordinate system, centered on the scene or a chosen object.

One of the most important features of the application integrating the evidence documentation should then be the possibility to bring into register the image data acquired with different imaging modalities. The spatial resolution of two scanning devices depends on their distances from the object. Finding the aligning transformation should be based on the point-to-surface distance rather then point-to-point distance as it is in classical Iterative Closest Point (ICP) algorithm [25] commonly used for this purpose. This approach requires the initial cloud points data to be triangulated.

Merging all the data into a single 3D model is neither feasible nor necessary. Like the technician at the scene, the expert in the lab can be given the ability to move around in a relatively low-resolution model of the entire scene, generated from one or more overview scan or photogrammetric reconstruction. While exploring this model, more detailed local representations can be called up. Other data may also be invoked from this level, such as text notes, measurements, and references to collected physical samples and artifacts.

An example of an integrating program was implemented by modifying the CloudCompare, where clicking on a label opens either Google Maps, a panoramic view, or a scan or photograph made from the indicated viewpoint (Figure 14 and Figure 15). Fusion of short- and long-range photogrammetric reconstructions is shown in Figure 1. Camera positions are indicated by color rectangles representing images. Close-up photographs of the dummy head (red rectangles) were taken on a table in the lab, and only later virtually placed in the scene.

The data must be transmitted to the server and stored there in their original form, complete with checksums or similar means of ensuring data integrity. The choice of objects to scan can be made by the field technician or by the support center, using two-way graphic communication.

One advantage of such joint presentation of the scene model is that it can be linked to the incident scene pilot program directing the on-scene data acquisition. The arrival of additional data means more information about a fragment of the scene, informing the investigators about what was found there and which areas need more exploration using a higher resolution or shorter range scanner. The pilot program will also determine which areas have not been scanned yet and help make an informed decision as to which of them should be imaged, by what techniques, and at what level of detail.

## 5. Facilitating Evidence Collection

An incident scene pilot application should facilitate the sharing of visual information in the form of images with a graphic overlay (of sketches and markers) applied on them. Its design should follow the following assumptions:

Several persons with mobile devices should be able to work with the same image simultaneously, and sketches drawn by one user should be visible to all; the server should keep track of changes. The raw images should never be altered, and data transfer during field work should be kept to a minimum.

The Sketchbook (Figure 16) is a simple internet application run in a web browser, intended as a demonstration of the concept of an incident scene pilot. Being implemented as a web page, the application can be used on any device and with any operating system, as long as HTML5 and JavaScript are supported.

The server part of the application, written in PHP, handles the transfer of data from/to the database. The client application calls it via JQuery messages. The drawings and markers added by the user are stored in the database with a time stamp and user ID. Every operation sent to the database is related to a particular photograph, making it possible to work with several photos or spherical panoramas in parallel.

## 6. Conclusions

Digitizing an incident scene for forensic purposes with the use of modern 2D and 3D image acquisition devices requires software tools capable of handling and integrating all the various data formats. A comparative analysis of the accuracy and precision of scanners and of photogrammetric reconstructions suggests a hierarchic approach to the digitization process. Views can be generated at various levels, from a general overview of the scene to detailed close-ups of traces and other pieces of evidence, within the limits of available data.

Integrating 2D images with 3D is possible by registering the photogrammetric reconstructions to 3D scans.

According to Table 2 and Table 3, the precision of photogrammetric methods at close-range is in the same class as laser scanning. Middle-range photogrammetry is comparable to the less precise among TLS scanners. Only far-range photogrammetry is still clearly inferior to 3D scanning.

It is usually impossible to build a complete model of the scene using the most precise devices, because of the scanning time and data volume it would involve. Therefore, managing the process of data acquisition is a series of tradeoffs between the potential of the available devices, the requirements of the investigation, and the cost of acquiring, transferring, processing, and storing data.

This is a task for a process (or a person) managing the digital acquisition of the scene: deciding which areas require another scan, and which details should be scanned at a greater resolution. The Sketchbook program can be used to close the loop between the supporting expert and the field technician.

As machine learning continues to develop, AI solutions can be expected to replace increasingly sophisticated parts of the work performed by humans. One of the tasks which can thus be automated is annotating sketches by labeling the objects found in them and deciding which distances (and possibly angles, areas, and volumes) should be explicitly placed on the sketch, in order to make it a comprehensive summary of the collected evidence.

## Figures and Tables

**Figure 1 sensors-21-01407-f001:**
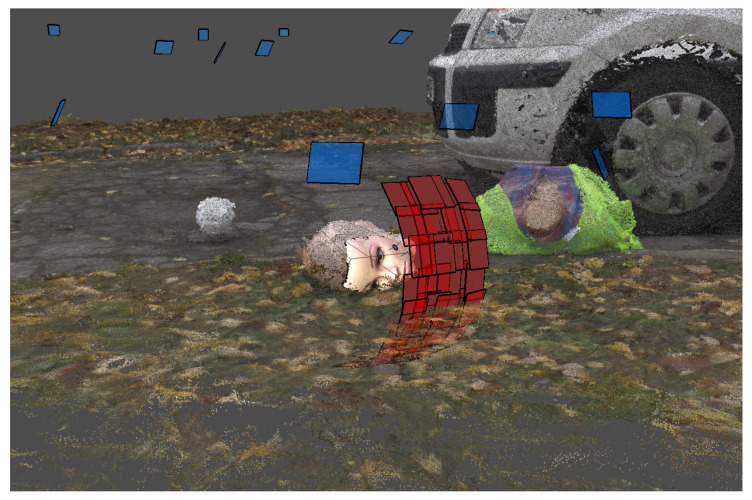
Fusion of short- and long-range photogrammetry.

**Figure 2 sensors-21-01407-f002:**
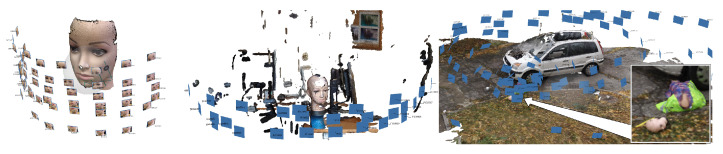
Photogrammetric reconstruction for close-range, middle-range, and long-range photography.

**Figure 3 sensors-21-01407-f003:**
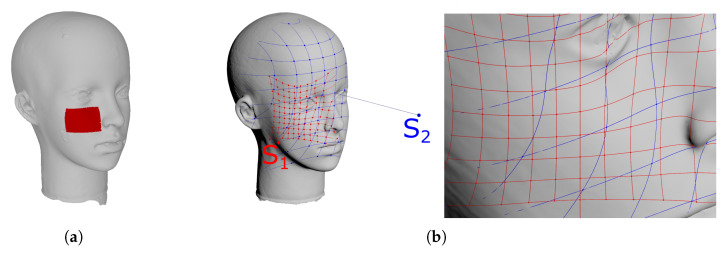
The fragment of the reference object analyzed in preliminary tests (**a**).The influence of the camera distance and direction on the spatial resolution of measurements (**b**).

**Figure 4 sensors-21-01407-f004:**
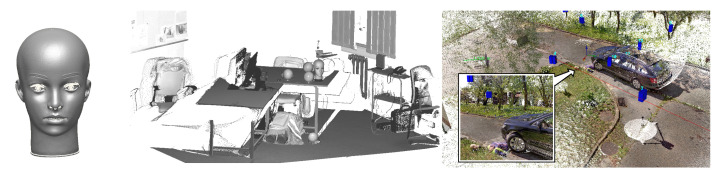
3D scans of dummy face: reference Konica Minolta scan, scan in the lab, overview of outside scans.

**Figure 5 sensors-21-01407-f005:**
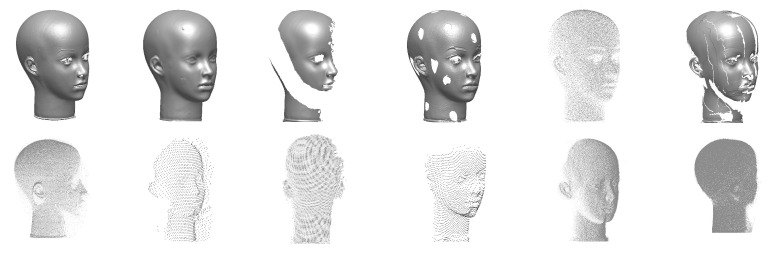
Close-up scans of the dummy head (top), left to right: Konica Minolta VI-9i (tele), MetraSCAN 210, Go!Scan 20, Konica Minolta VI-9i (middle),Faro Freestyle3D, Faro ScanArm. All represented by triangular meshes except FreeStyle (point cloud). Remote scans (bottom), left to right: Z+F 5010C (detail scan), Z+F 5010C (overall scan), Faro X 130 (detail scan), Faro X 130 (overall scan), Faro LS880, Trimble TX8, all as point clouds.

**Figure 6 sensors-21-01407-f006:**
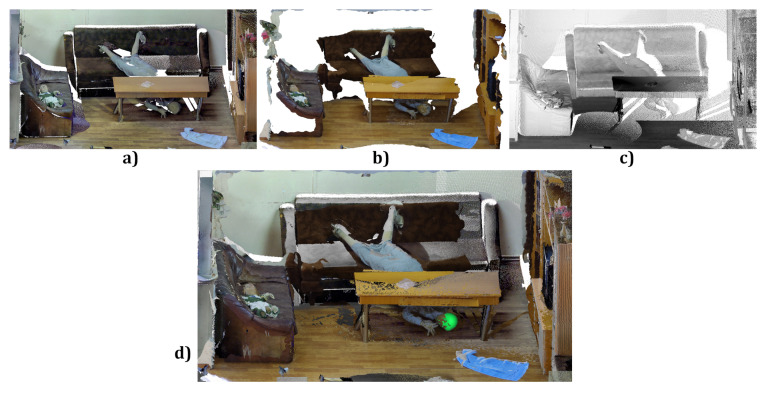
Faro scan (**a**), Z+F scan (**b**), photogrammetry (**c**) and their superposition (**d**).

**Figure 7 sensors-21-01407-f007:**
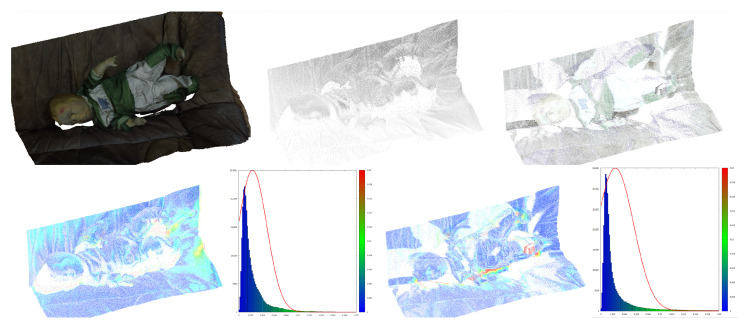
Top, left to right: middle-range photogrammetry, Faro scan, Z+F scan. Bottom left to right: Faro vs. photogrammetry (distance distribution and histogram, mean dist = 2.36 mm, stdev = 2.43 mm), Z+F vs. photogrammetry (distance distribution and histogram, mean dist = 2.48 mm, stdev = 3.09 mm).

**Figure 8 sensors-21-01407-f008:**
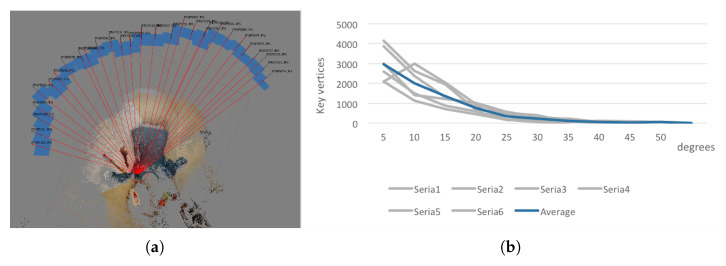
Determining the angular distance between photographs (**a**). The influence of angular distance between observations on the number of key vertices (**b**).

**Figure 9 sensors-21-01407-f009:**
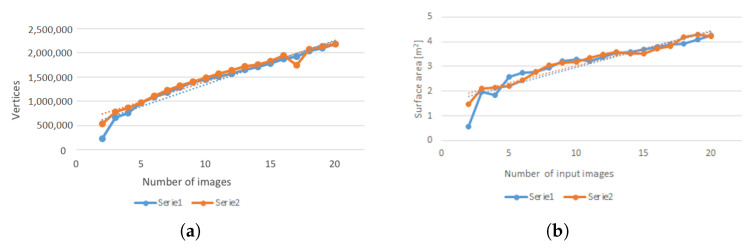
The influence of the number of input images on the number of vertices in the reconstruction (**a**), and on the surface area after reconstruction (**b**).

**Figure 11 sensors-21-01407-f011:**
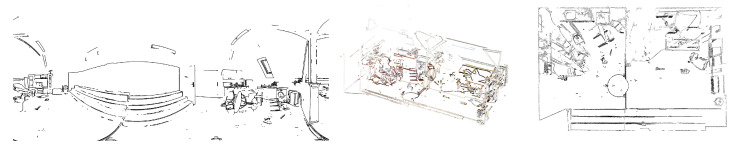
Edges detected in a 2D image (**left**), a single 3D scan after removing the points not lying on edges detected in the 2D image (**center**), and a sketch automatically generated from a 3D scan (**right**).

**Figure 12 sensors-21-01407-f012:**
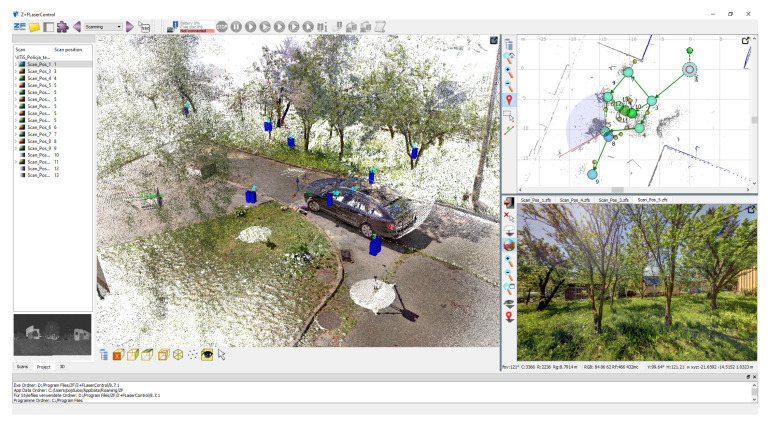
Z+F Laser Control software.

**Figure 13 sensors-21-01407-f013:**
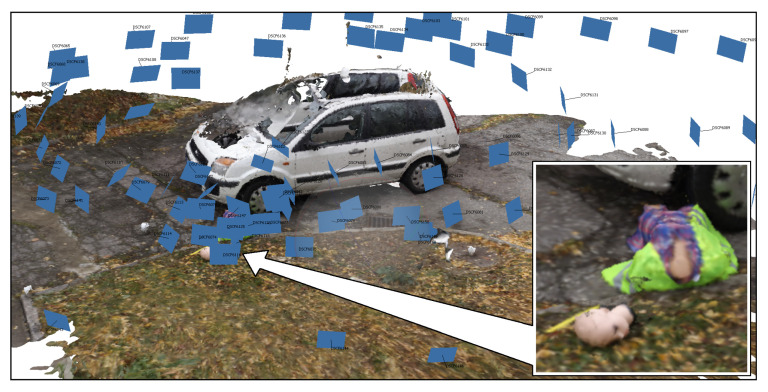
Agisoft Metashape software.

**Figure 14 sensors-21-01407-f014:**
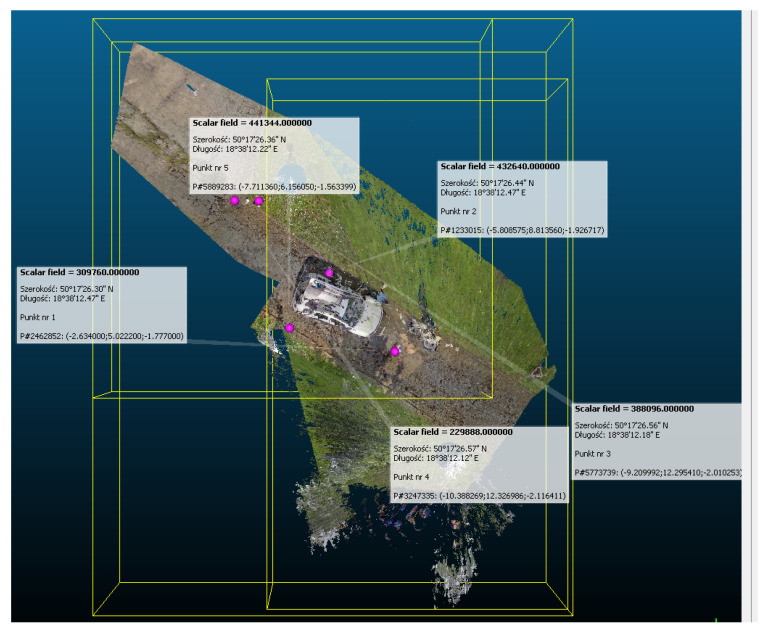
The location from GNSS pointed on the scan using our IncidentViewer (modification of CloudCompare).

**Figure 15 sensors-21-01407-f015:**
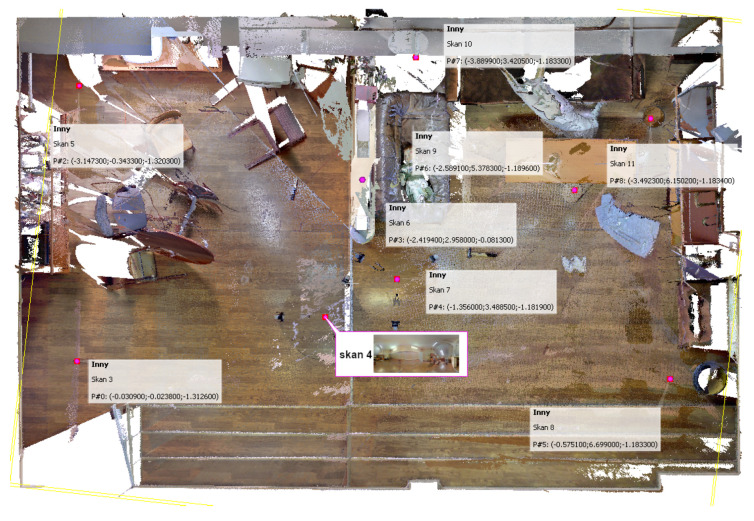
Integrating scans taken from different viewpoints—the viewpoints are shown on a floor-plan orthogonal projection of the scene model.

**Figure 16 sensors-21-01407-f016:**
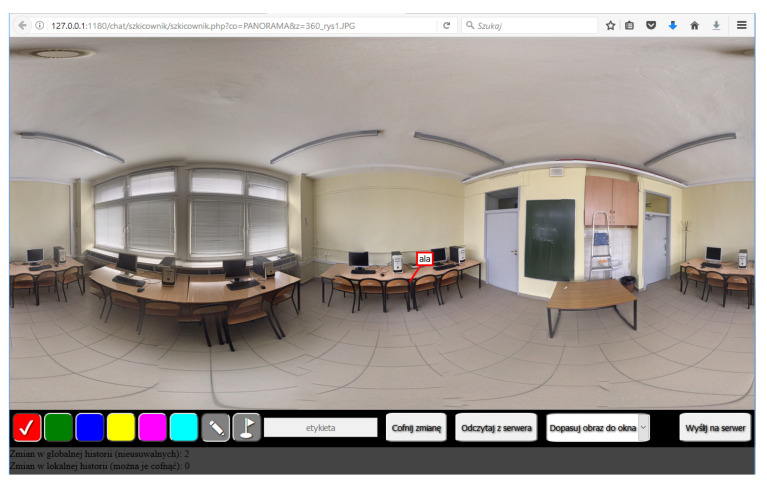
General view of the Sketchbook application.

**Table 1 sensors-21-01407-t001:** Preliminary comparison of the fragment of the MetraScan data to Konica Minolta VI-9i tele lens reference scan.

Method	Mean Dist	Stdev
EDMA	0.645	0.722
C2C	0.142	0.076
C2M	0.071	0.083

**Table 2 sensors-21-01407-t002:** Accuracy of photogrammetric 3D reconstructions of Figure 2.

Photography	Photogramm. to Scan		
	Mean Dist mm	Stdev mm	Points per cm^2^
close-range	0.22	0.487	1190
middle-range	1.72	1.66	50.4
long-range	8.89	9.75	2.2

**Table 3 sensors-21-01407-t003:** Estimated accuracy of scanners.

Scanner	Mean Dist mm	Stdev mm	Points per cm^2^
Close-Range scanners			
MetraSCAN 210	0.09	0.18	86.4
Go!Scan 20	0.11	0.26	91.4
Konica Minolta VI-9i (middle)	0.11	0.3	264
Faro Freestyle3D	0.32	0.37	118
Faro ScanArm	0.68	0.49	1223
TLS scanners			
Z+F 5010C (detail scan)	0.40	0.35	156
Z+F 5010C (overall scan)	0.53	0.97	3.7
Faro X 130 (detail scan)	0.78	0.85	34.3
Faro X 130 (overall scan)	1.05	1.55	4.5
Faro LS880	1.61	1.64	105.5
Trimble TX8	1.82	1.59	119.2

## Data Availability

The data presented in this study are openly available in Zenodo at doi:10.5281/zenodo.4544275.

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
