# Peer review of "Multimodal Imagery in Forensic Incident Scene Documentation"

_sensors, 2021, doi:10.3390/s21041407_

Round 1

Reviewer 1 Report

This paper presents an interesting topic in scene documentation. The authors evaluated various imaging modalities or use in forensic incident (crime or accident) scene documentation. Finally, the authors provide insights into their findings. This is an interesting paper. However, at this point, I would like to point out some issues to improve the current version of this paper. 

1. It is not clear what is the technical contribution of the paper. I think the authors should make it clear in the introduction section.

2. Does the size of the data affect the scene documentation process? If yes how? Perhaps providing some evaluation on how different datasets could affect the documentation process could help you provide some guidelines for those interesting in such techniques.

3. Although the paper is dealing with scene documentation, I think there should be a linkage with machine learning automatic techniques that perform summarization. Perhaps a future direction of this project is to use knowledge from summarization-related techniques and apply them to scene documentation. Example papers are the followings:

-- Generative Adversarial Network with Policy Gradient for Text Summarization

-- Pretraining-Based Natural Language Generation for Text Summarization

Overall, this paper has some research contributions, however, at this point, I recommend major revision. I feel that after addressing the mentioned issues the paper will be ready for publication.

Reviewer 2 Report

This manuscript presents a photogrammetric approach and methodology of reconstructing 3D images using 2D images. 

The methods, instrumentation (acquisition devices) and software tools to integrate the images are described in very good detail and would interest the readers. The introduction is well-written and the description of the processes involved are very adequate. Unfortunately no measurements were presented to assess the methods described.

The author(s) should present a measured quantity (accuracy of reconstructed image) as a metric to assess the complete methodology proposed. A comparison of those figures with others in previous literature will be very useful as well.

As it stands, this article is only an account of the photogrammetric approach, without any results to be drawn.

Please, revise and provide more detailed results about your experimental work.

Reviewer 3 Report

I think the authors addressed the reviewers' comments properly and improved the manuscript significantly. I recommend it for publication.

Author Response

Dear Reviewer 3,

thank you for your remarks.

As you did not require any modifications in our manuscript, we gratefully take your review as an approval.

With best regads,

Leszek Luchowski et al.

Round 2

Reviewer 1 Report

The authors were able to address all raised issues. For this reason, I would like to recommend this paper for the Sensors Journal.

Reviewer 2 Report

The revised manuscript is complete and of a great significance and quality.

Note to the authors:

Thank you for addressing my comments from the previous review.